# Do Neural Optimal Transport Solvers Work?
# A Continuous Wasserstein-2 Benchmark

**Alexander Korotin**
Skolkovo Institute of Science and Technology
Artificial Intelligence Research Institute
*Moscow, Russia*
`a.korotin@skoltech.ru`

**Lingxiao Li**
Massachusetts Institute of Technology
*Cambridge, Massachusetts, USA*
`lingxiao@mit.edu`

**Aude Genevay**
Massachusetts Institute of Technology
*Cambridge, Massachusetts, USA*
`aude.genevay@gmail.com`

**Justin Solomon**
Massachusetts Institute of Technology
*Cambridge, Massachusetts, USA*
`jsolomon@mit.edu`

**Alexander Filippov**
Huawei Noah's Ark Lab
*Moscow, Russia*
`filippov.alexander@huawei.com`

**Evgeny Burnaev**
Skolkovo Institute of Science and Technology
Artificial Intelligence Research Institute
*Moscow, Russia*
`e.burnaev@skoltech.ru`

## Abstract

Despite the recent popularity of neural network-based solvers for optimal transport (OT), there is no standard quantitative way to evaluate their performance. In this paper, we address this issue for quadratic-cost transport—specifically, computation of the Wasserstein-2 distance, a commonly-used formulation of optimal transport in machine learning. To overcome the challenge of computing ground truth transport maps between continuous measures needed to assess these solvers, we use input-convex neural networks (ICNN) to construct pairs of measures whose ground truth OT maps can be obtained analytically. This strategy yields pairs of continuous benchmark measures in high-dimensional spaces such as spaces of images. We thoroughly evaluate existing optimal transport solvers using these benchmark measures. Even though these solvers perform well in downstream tasks, many do not faithfully recover optimal transport maps. To investigate the cause of this discrepancy, we further test the solvers in a setting of image generation. Our study reveals crucial limitations of existing solvers and shows that increased OT accuracy does not necessarily correlate to better results downstream.

Solving optimal transport (OT) with continuous methods has become widespread in machine learning, including methods for large-scale OT [11, 36] and the popular Wasserstein Generative Adversarial Network (W-GAN) [3, 12]. Rather than discretizing the problem [31], continuous OT algorithms use neural networks or kernel expansions to estimate transport maps or dual solutions. This helps scale OT to large-scale and higher-dimensional problems not handled by discrete methods. Notable successes of continuous OT are in generative modeling [42, 20, 19, 7] and domain adaptation [43, 37, 25].

In these applications, OT is typically incorporated as part of the loss terms for a neural network model. For example, in W-GANs, the OT cost is used as a loss function for the generator; the model incorporates a neural network-based OT solver to estimate the loss. Although recent W-GANs provide state-of-the-art generative performance, however, it remains unclear to which extent this

success is connected to OT. For example, [28, 32, 38] show that popular solvers for the Wasserstein-1 ($\mathbb{W}_1$) distance in GANs fail to estimate $\mathbb{W}_1$ accurately. While W-GANs were initially introduced with $\mathbb{W}_1$ in [3], state-of-the art solvers now use both $\mathbb{W}_1$ and $\mathbb{W}_2$ (the *Wasserstein-2* distance, i.e., OT with the quadratic cost). While their experimental performance on GANs is similar, $\mathbb{W}_2$ solvers tend to converge faster (see [19, Table 4]) with better theoretical guarantees [19, 26, 16].

**Contributions.** In this paper, we develop a generic methodology for evaluating continuous quadratic-cost OT solvers ($\mathbb{W}_2$). Our main contributions are as follows:

- We use input-convex neural networks (ICNNs [2]) to construct pairs of continuous measures that we use as a benchmark with analytically-known solutions for quadratic-cost OT (§3, §4.1).
- We use these benchmark measures to evaluate popular quadratic-cost OT solvers in high-dimensional spaces (§4.3), including the image space of $64 \times 64$ CelebA faces (§4.4).
- We evaluate the performance of these OT solvers as a loss in generative modeling of images (§4.5).

Our experiments show that some OT solvers exhibit moderate error even in small dimensions (§4.3), performing similarly to trivial baselines (§4.2). The most successful solvers are those using parametrization via ICNNs. Surprisingly, however, solvers that faithfully recover $\mathbb{W}_2$ maps across dimensions struggle to achieve state-of-the-art performance in generative modeling.

Our benchmark measures can be used to evaluate future $\mathbb{W}_2$ solvers in high-dimensional spaces, a crucial step to improve the transparency and replicability of continuous OT research. Note the benchmark from [35] does not fulfill this purpose, since it is designed to test discrete OT methods and uses discrete low-dimensional measures with limited support.

**Notation.** We use $\mathcal{P}_2(\mathbb{R}^D)$ to denote the set of Borel probability measures on $\mathbb{R}^D$ with finite second moment and $\mathcal{P}_{2,ac}(\mathbb{R}^D)$ to denote its subset of absolutely continuous probability measures. We denote by $\Pi(\mathbb{P}, \mathbb{Q})$ the set of the set of probability measures on $\mathbb{R}^D \times \mathbb{R}^D$ with marginals $\mathbb{P}$ and $\mathbb{Q}$. For some measurable map $T : \mathbb{R}^D \to \mathbb{R}^D$, we denote by $T\sharp$ the associated push-forward operator. For $\phi : \mathbb{R}^D \to \mathbb{R}$, we denote by $\overline{\phi}$ its Legendre-Fenchel transform [10] defined by $\overline{\phi}(y) = \max_{x \in \mathbb{R}^D}[\langle x, y \rangle - \phi(x)]$. Recall that $\overline{\phi}$ is a convex function, even when $\phi$ is not.

# 1   Background on Optimal Transport

We start by stating the definition and some properties of optimal transport with quadratic cost. We refer the reader to [34, Chapter 1] for formal statements and proofs.

**Primal formulation.** For $\mathbb{P}, \mathbb{Q} \in \mathcal{P}_2(\mathbb{R}^D)$, Monge's primal formulation of the squared *Wasserstein-2* distance, i.e., OT with quadratic cost, is given by

$$\mathbb{W}_2^2(\mathbb{P}, \mathbb{Q}) \stackrel{\text{def}}{=} \min_{T\sharp\mathbb{P}=\mathbb{Q}} \int_{\mathbb{R}^D} \frac{\|x - T(x)\|^2}{2} d\mathbb{P}(x), \tag{1}$$

where the minimum is taken over measurable functions (transport maps) $T : \mathbb{R}^D \to \mathbb{R}^D$ mapping $\mathbb{P}$ to $\mathbb{Q}$. The optimal $T^*$ is called the *optimal transport map* (OT map). Note that (1) is not symmetric, and this formulation does not allow for mass splitting, i.e., for some $\mathbb{P}, \mathbb{Q} \in \mathcal{P}_2(\mathbb{R}^D)$, there is no map $T$ that satisfies $T\sharp\mathbb{P} = \mathbb{Q}$. Thus, Kantorovich proposed the following relaxation [14]:

$$\mathbb{W}_2^2(\mathbb{P}, \mathbb{Q}) \stackrel{\text{def}}{=} \min_{\pi \in \Pi(\mathbb{P},\mathbb{Q})} \int_{\mathbb{R}^D \times \mathbb{R}^D} \frac{\|x - y\|^2}{2} d\pi(x, y), \tag{2}$$

where the minimum is taken over all transport plans $\pi$, i.e., measures on $\mathbb{R}^D \times \mathbb{R}^D$ whose marginals are $\mathbb{P}$ and $\mathbb{Q}$. The optimal $\pi^* \in \Pi(\mathbb{P}, \mathbb{Q})$ is called the *optimal transport plan* (OT plan). If $\pi^*$ is of the form $[\text{id}_{\mathbb{R}^D}, T^*]\sharp\mathbb{P} \in \Pi(\mathbb{P}, \mathbb{Q})$ for some $T^*$, then $T^*$ is the minimizer of (1).

**Dual formulation.** For $\mathbb{P}, \mathbb{Q} \in \mathcal{P}_2(\mathbb{R}^D)$, the *dual formulation* of $\mathbb{W}_2^2$ is given by [40]:

$$\mathbb{W}_2^2(\mathbb{P}, \mathbb{Q}) = \max_{f \oplus g \leqslant \frac{1}{2}\|\cdot\|^2} \left[ \int_{\mathbb{R}^D} f(x)d\mathbb{P}(x) + \int_{\mathbb{R}^D} g(y)d\mathbb{Q}(y) \right], \tag{3}$$

where the maximum is taken over all $f \in \mathcal{L}^1(\mathbb{P}, \mathbb{R}^D \to \mathbb{R})$ and $g \in \mathcal{L}^1(\mathbb{Q}, \mathbb{R}^D \to \mathbb{R})$ satisfying $f(x) + g(y) \leqslant \frac{1}{2}\|x - y\|^2$ for all $x, y \in \mathbb{R}^D$. From the optimal dual potential $f^*$, we can recover the optimal transport plan $T^*(x) = x - \nabla f^*(x)$ [34, Theorem 1.17].

The optimal $f^*, g^*$ satisfy $(f^*)^c = g^*$ and $(g^*)^c = f^*$, where $u^c : \mathbb{R}^D \to \mathbb{R}$ is the $c-$transform of $u$ defined by $u^c(y) = \min_{x \in \mathbb{R}^D} \left[ \frac{1}{2} \|x - y\|^2 - u(x) \right]$. We can rewrite (3) as

$$\mathbb{W}_2^2(\mathbb{P}, \mathbb{Q}) = \max_f \left[ \int_{\mathbb{R}^D} f(x) d\mathbb{P}(x) + \int_{\mathbb{R}^D} f^c(y) d\mathbb{Q}(y) \right], \tag{4}$$

where the maximum is taken over all $f \in \mathcal{L}^1(\mathbb{P}, \mathbb{R}^D \to \mathbb{R})$. Since $f^*$ and $g^*$ are each other's $c$-transforms, they are both $c$-concave [34, §1.6], which is equivalent to saying that functions $\psi^* : x \mapsto \frac{1}{2} \|x\|^2 - f^*(x)$ and $\phi^* : x \mapsto \frac{1}{2} \|x\|^2 - g^*(x)$ are convex [34, Proposition 1.21]. In particular, $\overline{\psi^*} = \phi^*$ and $\overline{\phi^*} = \psi^*$. Since

$$T^*(x) = x - \nabla f^*(x) = \nabla \left( \frac{\|x\|^2}{2} - f^*(x) \right) = \nabla \psi^*, \tag{5}$$

we see that the OT maps are gradients of convex functions, a fact known as Brenier's theorem [6].

**"Solving" optimal transport problems.** In applications, for given $\mathbb{P}, \mathbb{Q} \in \mathcal{P}_2(\mathbb{R}^D)$, the $\mathbb{W}_2$ optimal transport problem is typically considered in the following three similar but not equivalent tasks:

- **Evaluating $\mathbb{W}_2^2(\mathbb{P}, \mathbb{Q})$.** The Wasserstein-2 distance is a geometrically meaningful way to compare probability measures, providing a metric on $\mathcal{P}_2(\mathbb{R}^D)$.
- **Computing the optimal map $T^*$ or plan $\pi^*$.** The map $T^*$ provides an intuitive way to interpolate between measures. It is often used as a generative map between measures in problems like domain adaptation [36, 43] and image style transfer [16].
- **Using the gradient $\partial \mathbb{W}_2^2(\mathbb{P}_\alpha, \mathbb{Q})/\partial \alpha$ to update generative models.** Derivatives of $\mathbb{W}_2^2$ are used implicitly in generative modeling that incorporates $\mathbb{W}_2$ loss [19, 33], in which case $\mathbb{P} = \mathbb{P}_\alpha$ is a parametric measure and $\mathbb{Q}$ is the data measure. Typically, $\mathbb{P}_\alpha = G_\alpha \sharp \mathbb{S}$ is the measure generated from a fixed latent measure $\mathbb{S}$ by a parameterized function $G_\alpha$, e.g., a neural network. The goal is to find parameters $\alpha$ that minimize $\mathbb{W}_2^2(\mathbb{P}_\alpha, \mathbb{Q})$ via gradient descent.

In the generative model setting, by definition of the pushforward $\mathbb{P}_\alpha = G_\alpha \sharp \mathbb{S}$, we have

$$\mathbb{W}_2^2(\mathbb{P}_\alpha, \mathbb{Q}) = \int_z f^*(G_\alpha(z)) d\mathbb{S}(z) + \int_{\mathbb{R}^D} g^*(y) d\mathbb{Q}(y),$$

where $f^*$ and $g^*$ are the optimal dual potentials. At each generator training step, $f^*$ and $g^*$ are fixed so that when we take the gradient with respect to $\alpha$, by applying the chain rule we have:

$$\frac{\partial \mathbb{W}_2^2(\mathbb{P}_\alpha, \mathbb{Q})}{\partial \alpha} = \int_z \mathbf{J}_\alpha G_\alpha(z)^T \nabla f^* \big( G_\alpha(z) \big) d\mathbb{S}(z), \tag{6}$$

where $\mathbf{J}_\alpha G_\alpha(z)^T$ is the transpose of the Jacobian matrix of $G_\alpha(z)$ w.r.t. parameters $\alpha$. This result still holds without assuming the potentials are fixed by the envelope theorem [29]. To capture the gradient, we need a good estimate of $\nabla f^* = \mathrm{id}_{\mathbb{R}^D} - T^*$ by (5). This task is somewhat different from computing the OT map $T^*$: since the estimate of $\nabla f^*$ is only involved in the gradient update for the generator, it is allowed to differ while still resulting in a good generative model.

We will use the generic phrase *OT solver* to refer to a method for solving any of the tasks above.

**Quantitative evaluation of OT solvers.** For discrete OT methods, a benchmark dataset [35] exists but the mechanism for producing the dataset does not extend to continuous OT. Existing continuous solvers are typically evaluated on a set of self-generated examples or tested in generative models without evaluating its actual OT performance. Two kinds of metrics are often used:

*Direct metrics* compare the computed transport map $\hat{T}$ with the true one $T^*$, e.g., by using $\mathcal{L}^2$ Unexplained Variance Percentage ($\mathcal{L}^2$-UVP) metric [16, §5.1], [17, §5]. There are relatively few direct metrics available, since the number of examples of $\mathbb{P}, \mathbb{Q}$ with known ground truth $T^*$ is small: it is known that $T^*$ can be analytically derived or explicitly computed in the discrete case [31, §3], 1-dimensional case [31, §2.6], and Gaussian/location-scatter cases [1].

*Indirect metrics* use an OT solver as a component in a larger pipeline, using end-to-end performance as a proxy for solver quality. For example, in generative modeling where OT is used as the generator loss [19, 27], the quality of the generator can be assessed through metrics for GANs, such as the Fréchet Inception distance (FID) [13]. Indirect metrics do not provide clear understanding about the quality of the solver itself, since they depend on components of the model that are not related to OT.

## 2 Continuous Dual Solvers for Quadratic Cost Transport

While our benchmark might be used to test any continuous solver which computes map $T^*$ or gradient $\nabla f^*$, in this paper, we perform evaluation only on *dual-form* continuous solvers based on (3) or (4). Such solvers have straightforward optimization procedures and can be adapted to various datasets without extensive hyperparameter search. In contrast, *primal-form* solvers based on (1), e.g., [18, 43, 21, 23], typically parameterize $T^*$ using complicated generative modeling techniques that depend on careful hyperparameter search and complex optimization procedures [24].

We summarize existing continuous dual form solvers in Table 1. These fit a parametric function $f_\theta$ (or $\psi_\theta$) to approximate $f^*$ (or $\psi^* = \mathrm{id}_{\mathbb{R}^D} - f^*$). The resulting $f_\theta$ produces an approximate OT map $\mathrm{id}_{\mathbb{R}^D} - \nabla f_\theta = \nabla \psi_\theta \approx T^*$ and derivative $\nabla f_\theta = \mathrm{id}_{\mathbb{R}^D} - \nabla \psi_\theta$ needed to update generative models (6).

| Solver | Related works | Parameterization of potentials or maps | Quantitatively tested as OT | Tested in GANs |
|---|---|---|---|---|
| Regularized [LS] | [11, 36, 33] | $f_\theta, g_\omega : \mathbb{R}^D \to \mathbb{R}$ - NNs | Gaussian case [16] | Ent.-regularized WGAN [33] |
| Maximin [MM] | [30] | $f_\theta : \mathbb{R}^D \to \mathbb{R}$ - NN $H_\omega : \mathbb{R}^D \to \mathbb{R}^D$ - NN | ✗ | Three-player WGAN [30] |
| Maximin (Batch-wise) [MM-B] | [27, 8] | $f_\theta : \mathbb{R}^D \to \mathbb{R}$ - NN | ✗ | (q,p)-WGAN [27] |
| Quadratic Cost [QC] | [19] | $f_\theta : \mathbb{R}^D \to \mathbb{R}$ - NN | ✗ | WGAN-QC [19] |
| Maximin + ICNN [MMv1] | [39] | $\psi_\theta : \mathbb{R}^D \to \mathbb{R}$ - ICNN | Gaussian case [16] | ✗ |
| Maximin + 2 ICNNs [MMv2] | [26, 9] | $\psi_\theta : \mathbb{R}^D \to \mathbb{R}$ - ICNN $H_\omega : \mathbb{R}^D \to \mathbb{R}^D$ - $\nabla$ICNN | Gaussian case [16] | ✗ |
| Non-Maximin [W2] | [16, 17] | $\psi_\theta : \mathbb{R}^D \to \mathbb{R}$ - ICNN $H_\omega : \mathbb{R}^D \to \mathbb{R}^D$ - $\nabla$ICNN | Gaussian case [16] | ✗ |

Table 1: Comprehensive table of existing continuous dual solvers for OT with the quadratic cost.

To our knowledge, none of these solvers has been quantitatively evaluated in a non-Gaussian setting. For [MM], [MM-B], and [QC], the quality of the recovered derivatives $\nabla f^*$ for $\partial \mathbb{W}_2^2(\mathbb{P}_\alpha, \mathbb{Q})/\partial \alpha$ has only been evaluated implicitly through GAN metrics. Moreover, these three solvers have not been quantitatively evaluated on solving OT tasks. We now overview each solver from Table 1.

[LS] optimizes an unconstrained regularized dual form of (3) [36]:

$$\max_{f,g} \left[ \int_{\mathbb{R}^D} f(x) d\mathbb{P}(x) + \int_{\mathbb{R}^D} g(y) d\mathbb{Q}(y) \right] - \mathcal{R}(f, g). \tag{7}$$

The *entropic* or *quadratic* regularizer $\mathcal{R}$ penalizes potentials $f, g$ for violating the constraint $f \oplus g \leqslant \frac{1}{2} \| \cdot \|^2$ [36, §3]. In practice, $f = f_\theta$ and $g = g_\omega$ are linear combinations of kernel functions [11] or neural networks [36]. The parameters $\theta, \omega$ are obtained by applying stochastic gradient ascent (SGA) over random mini-batches sampled from $\mathbb{P}, \mathbb{Q}$.

Most other solvers are based on an expansion of (4):

$$\mathbb{W}_2^2(\mathbb{P}, \mathbb{Q}) = \max_f \int_{\mathbb{R}^D} f(x) d\mathbb{P}(x) + \int_{\mathbb{R}^D} \overbrace{\min_{x \in \mathbb{R}^D} \left[ \frac{1}{2} \|x - y\|^2 - f(x) \right]}^{= f^c(y)} d\mathbb{Q}(y). \tag{8}$$

The challenge of (8) is the inner minimization over $x \in \mathbb{R}^D$, i.e., evaluating $f^c(y)$. The main difference between existing solvers is the procedure used to solve this inner problem.

[MM-B] uses a neural network $f_\theta$ as the potential trained using mini-batch SGA [27]. To solve the inner problem, the authors restrict the minimization of $x$ to the current mini-batch from $\mathbb{P}$ instead of $\mathbb{R}^D$. The strategy is fast but leads to *overestimation* of the inner problem's solution since the minimum is taken over a restricted subset.

[MM-v1] exploits the property that $f^* = \frac{1}{2} \| \cdot \|^2 - \psi^*$, where $\psi^*$ is convex [39]. The authors parametrize $f_\theta = \frac{1}{2} \| \cdot \|^2 - \psi_\theta$, where $\psi_\theta$ is an input convex neural network (ICNN) [2]. Hence, for every $y \in \mathbb{R}^D$, the inner problem of (8) becomes convex in $x$. This problem can be solved using SGA to high precision, but doing so is computationally costly [16, §C.4].

[MM] uses a formulation equivalent to (8) [30]:

$$\mathbb{W}_2^2(\mathbb{P}, \mathbb{Q}) = \max_f \int_{\mathbb{R}^D} f(x) d\mathbb{P}(x) + \int_{\mathbb{R}^D} \min_H \left[ \frac{1}{2} \|H(y) - y\|^2 - f(H(y)) \right] d\mathbb{Q}(y), \tag{9}$$

where the minimization is performed over functions $H : \mathbb{R}^D \to \mathbb{R}^D$. The authors use neural networks $f_\theta$ and $H_\omega$ to parametrize the potential and the minimizer of the inner problem. To train $\theta, \omega$, the authors apply stochastic gradient ascent/descent (SGAD) over mini-batches from $\mathbb{P}, \mathbb{Q}$. [MM] is generic and can be modified to compute arbitrary transport costs and derivatives, not just $\mathbb{W}_2^2$, although the authors have tested only on the Wasserstein-1 ($\mathbb{W}_1$) distance.

Similarly to [MMv1], [MMv2] parametrizes $f_\theta = \frac{1}{2}\| \cdot \|^2 - \psi_\theta$, where $\psi_\theta$ is an ICNN [26]. For a fixed $f_\theta$, the optimal solution $H$ is given by $H = (\nabla\psi_\theta)^{-1}$ which is an inverse gradient of a convex function, so it is also a gradient of a convex function. Hence, the authors parametrize $H_\omega = \nabla\phi_\omega$, where $\phi_\omega$ is an ICNN, and use [MM] to fit $\theta, \omega$.

[W2] uses the same ICNN parametrization as [26] but introduces *cycle-consistency* regularization to avoid solving a maximin problem [16, §4].

Finally, we highlight the solver [QC] [19]. Similarly to [MM-B], a neural network $f_\theta$ is used as the potential. When each pair of mini-batches $\{x_n\}, \{y_n\}$ from $\mathbb{P}, \mathbb{Q}$ is sampled, the authors solve a *discrete* OT problem to obtain dual variables $\{f_n^*\}, \{g_n^*\}$, which are used to regress $f_\theta(x_n)$ onto $f_n^*$.

**Gradient deviation.** The solvers above optimize for potentials like $f_\theta$ (or $\psi_\theta$), but it is the gradient of $f_\theta$ (or $\psi_\theta$) that is used to recover the OT map via $T = x - \nabla f_\theta$. Even if $\|f - f^*\|_{\mathcal{L}^2(\mathbb{P})}^2$ is small, the difference $\|\nabla f_\theta - \nabla f^*\|_{\mathcal{L}^2(\mathbb{P})}^2$ may be arbitrarily large since $\nabla f_\theta$ is not directly involved in optimization process. We call this issue *gradient deviation*. This issue is only addressed formally for ICNN-based solvers [MMv1], [MMv2], [W2] [16, Theorem 4.1], [26, Theorem 3.6].

**Reversed solvers.** [MM], [MMv2], [W2] recover not only the forward OT map $\nabla\psi_\theta \approx \nabla\psi^* = T^*$, but also the inverse, given by $H_\omega \approx (T^*)^{-1} = (\nabla\psi^*)^{-1} = \nabla\overline{\psi^*}$, see [26, §3] or [16, §4.1]. These solvers are asymmetric in $\mathbb{P}, \mathbb{Q}$ and an alternative is to swap $\mathbb{P}$ and $\mathbb{Q}$ during training. We denote such *reversed* solvers by [MM:R], [MMv2:R], [W2:R]. In §4 we show that surprisingly [MM:R] works better in generative modeling than [MM].

## 3 Benchmarking OT Solvers

In this section, we develop a generic method to produce *benchmark pairs*, i.e., measures $(\mathbb{P}, \mathbb{Q})$ such that $\mathbb{Q} = T\sharp\mathbb{P}$ with sample access and an analytically known OT solution $T^*$ between them.

**Key idea.** Our method is based on the fact that for a differentiable convex function $\psi : \mathbb{R}^D \to \mathbb{R}$, its gradient $\nabla\psi$ is an *optimal* transport map between any $\mathbb{P} \in \mathcal{P}_{2,ac}(\mathbb{R}^D)$ and its pushforward $\nabla\psi\sharp\mathbb{P}$ by $\nabla\psi : \mathbb{R}^D \to \mathbb{R}^D$. This follows from Brenier's theorem [6], [41, Theorem 2.12]. Thus, for a continuous measure $\mathbb{P}$ with sample access and a *known* convex $\psi$, $(\mathbb{P}, \nabla\psi\sharp\mathbb{P})$ can be used as a benchmark pair. We sample from $\nabla\psi\sharp\mathbb{P}$ by drawing samples from $\mathbb{P}$ and pushing forward by $\nabla\psi$.

**Arbitrary pairs** $(\mathbb{P}, \mathbb{Q})$. It is difficult to compute the exact continuous OT solution for an arbitrary pair $(\mathbb{P}, \mathbb{Q})$. As a compromise, we compute an *approximate* transport map as the gradient of an ICNN using [W2]. That is, we find $\psi_\theta$ parameterized as an ICNN such that $\nabla\psi_\theta\sharp\mathbb{P} \approx \mathbb{Q}$. Then, the modified pair $(\mathbb{P}, \nabla\psi_\theta\sharp\mathbb{P})$ can be used to benchmark OT solvers. We choose [W2] because it exhibits good performance in higher dimensions, but other solvers can also be used so long as $\psi_\theta$ is convex. Because of the choice of [W2], subsequent evaluation might slightly favor ICNN-based methods.

**Extensions.** Convex functions can be modified to produce more benchmark pairs. If $\psi_1, \dots, \psi_N$ are convex, then $\sigma(\psi_1, \dots, \psi_N)$ is convex when $\sigma : \mathbb{R}^N \to \mathbb{R}$ is convex and monotone. For example, $c \cdot \psi_1 \ (c \geq 0)$, $\sum_n \psi_n$, $\max_n \psi_n$ are convex, and their gradients produce new benchmark pairs.

**Inversion.** If $\nabla\psi_\theta$ is bijective, then the inverse transport map for $(\mathbb{P}, \nabla\psi_\theta\sharp\mathbb{P})$ exists and is given by $(\nabla\psi_\theta)^{-1}$. For each $y \in \mathbb{R}^D$, the value $(\nabla\psi_\theta)^{-1}(y)$ can be obtained by solving a *convex* problem [39, §6], [16, §3]. All ICNNs $\psi_\theta$ we use have bijective gradients $\nabla\psi_\theta$, as detailed in Appendix B.1.

## 4 Benchmark Details and Results

We implement our benchmark in PyTorch and provide the pre-trained transport maps for all the benchmark pairs. The code is publicly available at

> https://github.com/iamalexkorotin/Wasserstein2Benchmark

The experiments are conducted on 4 GTX 1080ti GPUs and require about 100 hours of computation (per GPU). We provide implementation details in Appendix B.

### 4.1 Datasets

**High-dimensional measures.** We develop benchmark pairs to test whether the OT solvers can redistribute mass among modes of measures. For this purpose, we use Gaussian mixtures in dimensions $D = 2^1, 2^2, \ldots, 2^8$. In each dimension $D$, we consider a random mixture $\mathbb{P}$ of 3 Gaussians and two random mixtures $\mathbb{Q}_1, \mathbb{Q}_2$ of 10 Gaussians. We train approximate transport maps $\nabla \psi_i \sharp \mathbb{P} \approx \mathbb{Q}_i$ ($i = 1, 2$) using the $\lfloor$W2$\rfloor$ solver. Each potential is an ICNN with DenseICNN architecture [16, §B.2]. We create a benchmark pair via the half-sum of computed potentials $(\mathbb{P}, \frac{1}{2}(\nabla \psi_1 + \nabla \psi_2) \sharp \mathbb{P})$. The first measure $\mathbb{P}$ is a mixture of 3 Gaussians and the second is obtained by averaging potentials, which transforms it to approximate mixtures of 10 Gaussians. See Appendix A.1 and Figure 1 for details.

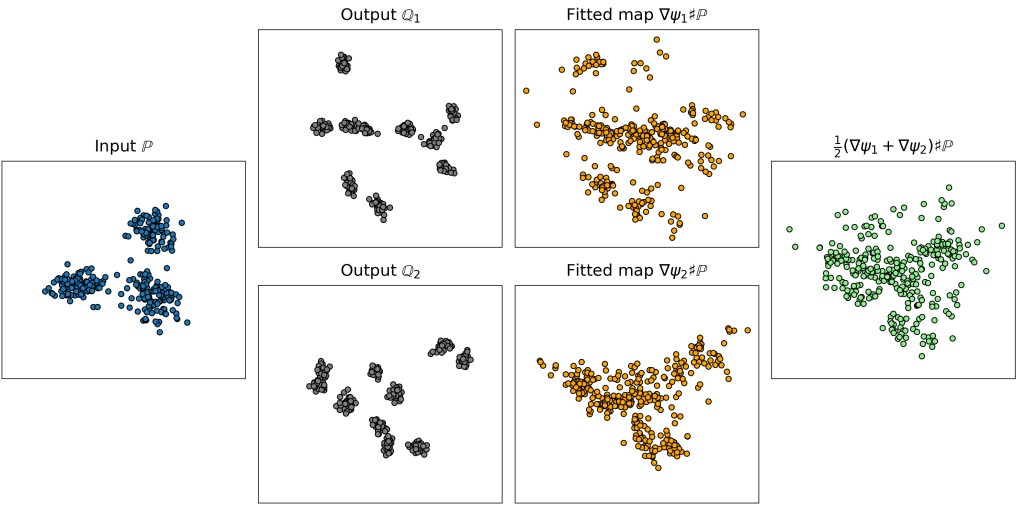

Figure 1: An example of creation of a benchmark pair for dimension $D = 16$. We first initialize 3 random Gaussian Mixtures $\mathbb{P}$ and $\mathbb{Q}_1, \mathbb{Q}_2$ and fit 2 approximate OT maps $\nabla \psi_i \sharp \mathbb{P} \approx \mathbb{Q}_i$, $i = 1, 2$. We use the average of potentials to define the output measure: $\frac{1}{2}(\nabla \psi_1 + \nabla \psi_2) \sharp \mathbb{P}$. Each scatter plot contains 512 random samples projected to 2 principle components of measure $\frac{1}{2}(\nabla \psi_1 + \nabla \psi_2) \sharp \mathbb{P}$.

**Images.** We use the aligned images of CelebA64 faces dataset[1] [22] to produce additional benchmark pairs. First, we fit 3 generative models (WGAN-QC [19]) on the dataset and pick intermediate training checkpoints to produce continuous measures $\mathbb{Q}^k_{\text{Early}}, \mathbb{Q}^k_{\text{Mid}}, \mathbb{Q}^k_{\text{Late}}$ for the first 2 models ($k = 1, 2$) and the final checkpoint of the third model ($k = 3$) to produce measure $\mathbb{P}^3_{\text{Final}}$. To make measures absolutely continuous, we add small Gaussian noise to the generator's output. Each checkpoint (Early, Mid, Late, Final) represents images of faces of a particular quality. Next, for $k \in \{1, 2\}$ and Cpkt $\in \{$Early, Mid, Late$\}$, we use $\lfloor$W2$\rfloor$ solver to fit an approximate transport map $\nabla \psi^k_{\text{Cpkt}}$ for the pair $(\mathbb{P}^3_{\text{Final}}, \mathbb{Q}^k_{\text{Cpkt}})$, i.e., $\nabla \psi^k_{\text{Cpkt}} \sharp \mathbb{P}^3_{\text{Final}} \approx \mathbb{Q}^k_{\text{Cpkt}}$. The potential $\psi^k_{\text{Cpkt}}$ is a convolutional ICNN with ConvICNN64 architecture (§B.1). For each Cpkt, we define a benchmark pair $(\mathbb{P}_{\text{CelebA}}, \mathbb{Q}_{\text{Cpkt}}) \stackrel{\text{def}}{=} (\mathbb{P}^3_{\text{Final}}, [(\nabla \psi^1_{\text{Cpkt}} + \nabla \psi^2_{\text{Cpkt}})/2] \sharp \mathbb{P}^3_{\text{Final}})$. See Appendix A.2 and Figure 2 for details.

### 4.2 Metrics and Baselines

**Baselines.** We propose three baseline methods: identity $\lfloor$ID$\rfloor$, constant $\lfloor$C$\rfloor$ and linear $\lfloor$L$\rfloor$. The *identity* solver outputs $T^{\text{id}} = \text{id}_{\mathbb{R}^D}$ as the transport map. The *constant* solver outputs the mean value of $\mathbb{Q}$, i.e., $T^0 \equiv \mathbb{E}_{\mathbb{Q}}[y] \equiv \mu_{\mathbb{Q}}$. The *linear* solver outputs $T^1(x) = \Sigma_{\mathbb{P}}^{-\frac{1}{2}} \left( \Sigma_{\mathbb{P}}^{\frac{1}{2}} \Sigma_{\mathbb{Q}} \Sigma_{\mathbb{P}}^{\frac{1}{2}} \right)^{\frac{1}{2}} \Sigma_{\mathbb{P}}^{-\frac{1}{2}} (x - \mu_{\mathbb{P}}) + \mu_{\mathbb{Q}}$, i.e., the OT map between measures coarsened to Gaussians [1, Theorem 2.3].

**Metrics.** To assess the quality of the recovered transport map $\hat{T} : \mathbb{R}^D \to \mathbb{R}^D$ from $\mathbb{P}$ to $\mathbb{Q}$, we use *unexplained variance percentage* (UVP) [16]: $\mathcal{L}^2$-UVP$(\hat{T}) \stackrel{\text{def}}{=} 100 \cdot \|\hat{T} - T^*\|^2_{\mathcal{L}^2(\mathbb{P})}/\text{Var}(\mathbb{Q})\%$. Here $T^*$ is the OT map. For values $\approx 0\%$, $\hat{T}$ approximates $T^*$ well. For values $\geqslant 100\%$, map $\hat{T}$ is far from optimal. The constant baseline provides $\mathcal{L}^2$-UVP$(T^0) = 100\%$.

---

[1] http://mmlab.ie.cuhk.edu.hk/projects/CelebA.html

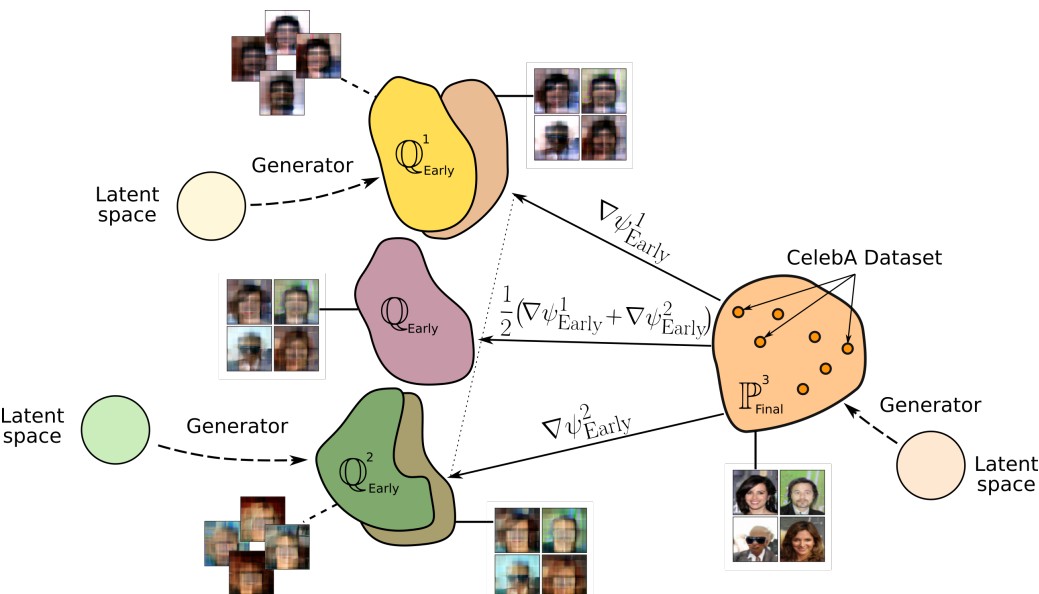

Figure 2: The pipeline of the image benchmark pair creation. We use 3 checkpoints of a generative model: $\mathbb{P}^3_{\text{Final}}$ (well-fitted) and $\mathbb{Q}^1_{\text{Cpkt}}$, $\mathbb{Q}^2_{\text{Cpkt}}$ (under-fitted). For $k = 1, 2$ we fit an approximate OT map $\mathbb{P}^3_{\text{Final}} \to \mathbb{Q}^k_{\text{Cpkt}}$ by $\nabla\psi^k_{\text{Cpkt}}$, i.e. a gradient of ICNN. We define the benchmark pair by $(\mathbb{P}_{\text{CelebA}}, \mathbb{Q}_{\text{Cpkt}}) \overset{def}{=} (\mathbb{P}^3_{\text{Final}}, \frac{1}{2}(\nabla\psi^1_{\text{Cpkt}} + \nabla\psi^2_{\text{Cpkt}})\sharp\mathbb{P}^3_{\text{Final}})$. In the visualization, Cpkt is Early.

To measure the quality of approximation of the derivative of the potential $[\text{id}_{\mathbb{R}^D} - \hat{T}] \approx \nabla f^*$ that is used to update generative models (6), we use *cosine similarity* (cos):

$$\cos(\text{id} - \hat{T}, \text{id} - T^*) \overset{\text{def}}{=} \frac{\langle \hat{T} - \text{id}, \nabla\psi^* - \text{id}\rangle_{\mathcal{L}^2(\mathbb{P})}}{\|T^* - \text{id}\|_{\mathcal{L}^2(\mathbb{P})} \cdot \|\hat{T} - \text{id}\|_{\mathcal{L}^2(\mathbb{P})}} \in [-1, 1].$$

To estimate $\mathcal{L}^2$-UVP and cos metrics, we use $2^{14}$ random samples from $\mathbb{P}$.

### 4.3 Evaluation of Solvers on High-dimensional Benchmark Pairs

We evaluate the solvers on the benchmark and report the computed metric values for the fitted transport map. For fair comparison, in each method the potential $f$ and the map $H$ (where applicable) are parametrized as $f_\theta = \frac{1}{2}\|\cdot\|^2 - \psi_\theta$ and $H_\omega = \nabla\phi_\omega$ respectively, where $\psi_\theta, \phi_\omega$ use DenseICNN architectures [16, §B.2]. In solvers $\lfloor QC\rfloor$, $\lfloor LS\rfloor$, $\lfloor MM\text{-}B\rfloor$, $\lfloor MM\rfloor$ we do not impose any restrictions on the weights $\theta, \omega$, i.e. $\psi_\theta, \phi_\omega$ are usual fully connected nets with additional skip connections. We provide the computed metric values in Table 2 and visualize fitted maps (for $D = 64$) in Figure 3.

All the solvers perform well ($\mathcal{L}^2$-UVP $\approx 0$, cos $\approx 1$) in dimension $D = 2$. In higher dimensions, only $\lfloor MMv1\rfloor$, $\lfloor MM\rfloor$, $\lfloor MMv2\rfloor$, $\lfloor W2\rfloor$ and their reversed versions produce reasonable results. However, $\lfloor MMv1\rfloor$ solver is slow since each optimization step solves a hard subproblem for computing $f^c$. Maximin solvers $\lfloor MM\rfloor$, $\lfloor MMv2\rfloor$, $\lfloor MM\text{:}R\rfloor$ are also hard to optimize: they either diverge from the start ($\rightarrowtail$) or diverge after converging to nearly-optimal saddle point ($\looparrowright$). This behavior is typical for maximin optimization and possibly can be avoided by a more careful choice of hyperparameters.

For $\lfloor QC\rfloor$, $\lfloor LS\rfloor$, $\lfloor MM\text{-}B\rfloor$, as the dimension increases, the $\mathcal{L}^2$-UVP drastically grows. Only $\lfloor MM\text{-}B\rfloor$ notably outperforms the trivial $\lfloor L\rfloor$ baseline. The error of $\lfloor MM\text{-}B\rfloor$ is explained by the overestimation of the inner problem in (8), yielding biased optimal potentials. The error of $\lfloor LS\rfloor$ comes from bias introduced by regularization [36]. In $\lfloor QC\rfloor$, error arises because a discrete OT problem solved on sampled mini-batches, which is typically biased [5, Theorem 1], is used to update $f_\theta$. Interestingly, although $\lfloor QC\rfloor$, $\lfloor LS\rfloor$ are imprecise in terms of $\mathcal{L}^2$-UVP, they provide a high cos metric.

Due to optimization issues and performance differences, wall-clock times for convergence are not representative. All solvers except $\lfloor MMv1\rfloor$ converged in several hours. Among solvers that

| Dim | 2 | 4 | 8 | 16 | 32 | 64 | 128 | 256 | Dim | 2 | 4 | 8 | 16 | 32 | 64 | 128 | 256 |
|---|---|---|---|---|---|---|---|---|---|---|---|---|---|---|---|---|---|
| [MMv1] | 0.2 | 1.0 | 1.8 | 1.4 | 6.9 | 8.1 | 2.2 | 2.6 | [MMv1] | 0.99 | 0.99 | 0.99 | 0.99 | 0.98 | 0.97 | 0.99 | 0.99 |
| [MM] | 0.1 | 0.3 | 0.9 | 2.2 | 4.2 | 3.2 | 3.1↷ | 4.1↷ | [MM] | 0.99 | 0.99 | 0.99 | 0.99 | 0.99 | 0.99 | 0.99↷ | 0.99↷ |
| [MM:R] | 0.1 | 0.3 | 0.7 | 1.9 | 2.8 | 4.5 | →↷ | →↷ | [MM:R] | 0.99 | 1.00 | 1.00 | 0.99 | 1.00 | 0.98 | →↷ | →↷ |
| [MMv2] | 0.1 | 0.68 | 2.2 | 3.1 | 5.3 | 10.1↷ | 3.2↷ | 2.7↷ | [MMv2] | 0.99 | 0.99 | 0.99 | 0.99 | 0.99 | 0.96↷ | 0.99↷ | 0.99↷ |
| [MMv2:R] | 0.1 | 0.7 | 4.4 | 7.7 | 5.8 | 6.8 | 2.1 | 2.8 | [MMv2:R] | 0.99 | 1.00 | 0.97 | 0.96 | 0.99 | 0.97 | 0.99 | 1.00 |
| [W2] | 0.1 | 0.7 | 2.6 | 3.3 | 6.0 | 7.2 | 2.0 | 2.7 | [W2] | 0.99 | 0.99 | 0.99 | 0.99 | 0.99 | 0.97 | 1.00 | 1.00 |
| [W2:R] | 0.2 | 0.9 | 4.0 | 5.3 | 5.2 | 7.0 | 2.0 | 2.7 | [W2:R] | 0.99 | 1.00 | 0.98 | 0.98 | 0.99 | 0.97 | 1.00 | 1.00 |
| [MM-B] | 0.1 | 0.7 | 3.1 | 6.4 | 12.0 | 13.9 | 19.0 | 22.5 | [MM-B] | 0.99 | 1.00 | 0.98 | 0.96 | 0.96 | 0.94 | 0.93 | 0.93 |
| [LS] | 5.0 | 11.6 | 21.5 | 31.7 | 42.1 | 40.1 | 46.8 | 54.7 | [LS] | 0.94 | 0.86 | 0.80 | 0.80 | 0.81 | 0.83 | 0.82 | 0.81 |
| [L] | 14.1 | 14.9 | 27.3 | 41.6 | 55.3 | 63.9 | 63.6 | 67.4 | [L] | 0.75 | 0.80 | 0.73 | 0.73 | 0.76 | 0.75 | 0.77 | 0.77 |
| [QC] | 1.5 | 14.5 | 28.6 | 47.2 | 64.0 | 75.2 | 80.5 | 88.2 | [QC] | 0.99 | 0.84 | 0.78 | 0.70 | 0.70 | 0.70 | 0.69 | 0.66 |
| [C] | 100 | 100 | 100 | 100 | 100 | 100 | 100 | 100 | [C] | 0.29 | 0.32 | 0.38 | 0.46 | 0.55 | 0.58 | 0.60 | 0.62 |
| [ID] | 32.7 | 42.0 | 58.6 | 87 | 121 | 137 | 145 | 153 | [ID] | 0.00 | 0.00 | 0.00 | 0.00 | 0.00 | 0.00 | 0.00 | 0.00 |

Table 2: $\mathcal{L}^2$-UVP (%, on the left) and $\cos \in [-1, 1]$ (on the right) metric values for transport maps fitted by OT solvers on the high-dimensional benchmark in dimensions $D = 2, 2^2, \ldots, 2^8$. Orange highlights $\mathcal{L}^2$-UVP > 10% and $\cos < 0.95$. Red indicates performance worse than [L] baseline.

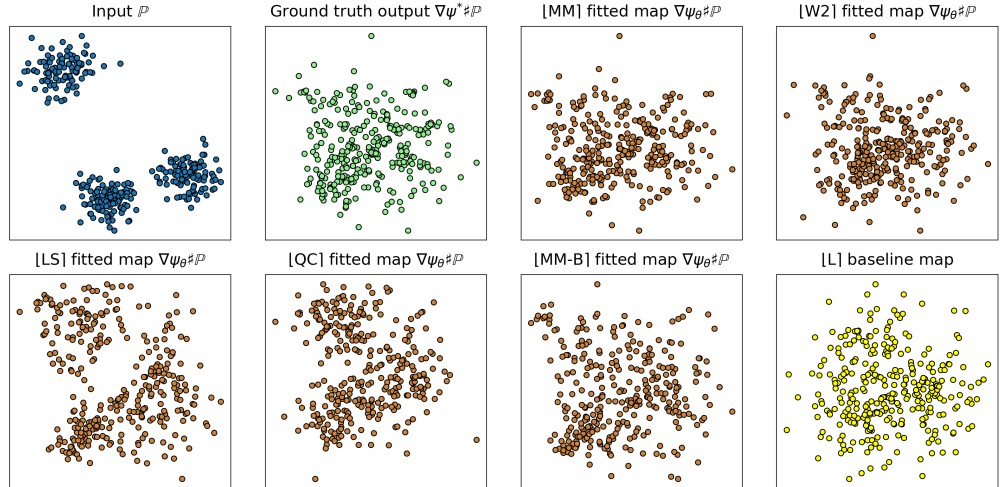

Figure 3: Visualization of a 64-dimensional benchmark pair and OT maps fitted by the solvers. Scatter plots contain 512 random samples projected onto 2 principal components of measure $\nabla\psi^* \sharp \mathbb{P}$.

substantially outperform the linear baseline, i.e. [MM], [MMv1], [MMv2], [W2], [MM-B], the fastest converging one is [MM-B], but it is biased. [MM], [MMv2], [W2] require more time.

## 4.4 Evaluation of Solvers in CelebA $64 \times 64$ Images Benchmark Pairs

For evaluation on the CelebA benchmark, we excluded [LS] and [MMv1]: the first is unstable in high dimensions [33], and the second takes too long to converge. ICNN-based solvers [MMv2], [W2] and their reversed versions perform roughly the same in this experiment. For simplicity, we treat them as one solver [W2].

In [W2], we parametrize $f_\theta = \frac{1}{2}\|\cdot\|^2 - \psi_\theta$ and $H_\omega = \nabla\phi_\omega$, where $\psi_\theta, \phi_\omega$ are input-convex neural nets with ConvexICNN64 architecture (§B.1). All the other solvers are designed in the generative modeling setting to work with convolutional architectures for images. Thus, in [MM], [QC], [MM-B] we parametrize networks $f_\theta$ as ResNet and $H_\omega$ as U-Net (in [MM]). In turn, in [MM:R] we parametrize $T_\theta$ by UNet and $g_\omega$ by ResNet.

We compute the transport map $\mathbb{Q}_{\text{Cpkt}} \to \mathbb{P}_{\text{CelebA}}$ for each solver on three image benchmarks. The results are in Figure 4 and Table 3 and echo patterns observed on high-dimensional problems (§4.3). [QC], [MM-B] suffer from extreme bias thanks to the high dimension of images, and the derivative of $\mathbb{W}_2^2$ computed by these solvers is almost orthogonal to the true derivative ($\cos \approx 0$). This means that *these solvers do not extract* $\mathbb{W}_2^2$. [MM], [MM:R], [W2] recover the transport maps well. [MM]'s map is slightly noisier than the one by [MM:R], a minor example of gradient deviation.

## 4.5 Evaluation of Solvers in Generative Modeling of CelebA $64 \times 64$ Faces

Based on our previous evaluation, many existing neural OT solvers are notably imprecise. This leads us to ask: *To what extent does solver quality matter in real-world applications?*

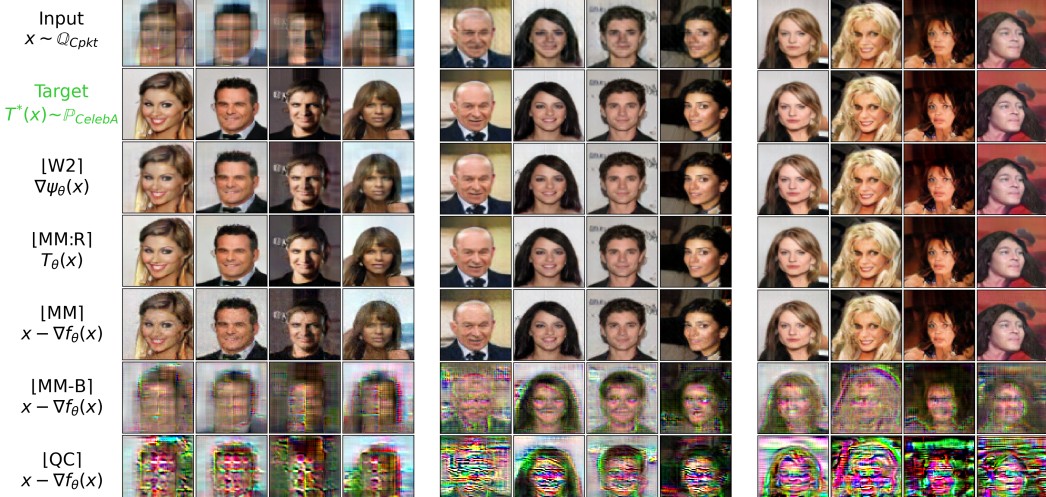

(a) Fitted maps $\mathbb{Q}_{\text{Early}} \to \mathbb{P}_{\text{CelebA}}$.     (b) Fitted maps $\mathbb{Q}_{\text{Mid}} \to \mathbb{P}_{\text{CelebA}}$. (c) Fitted maps $\mathbb{Q}_{\text{Late}} \to \mathbb{P}_{\text{CelebA}}$.

Figure 4: OT maps fitted by solvers on benchmarks $(\mathbb{Q}_{\text{Cpkt}}, \mathbb{P}_{\text{CelebA}})$. 1st line contains random $x \sim \mathbb{Q}_{\text{Cpkt}}$. 2nd line contains samples from $\mathbb{P}_{\text{CelebA}}$ obtained by pushing $x$ forward by OT map $T^* = \nabla\psi^*$. Subsequent lines show $x$ transported by maps fitted by OT solvers.

| Cpkt | Early | Mid | Late | | Cpkt | Early | Mid | Late |
|---|---|---|---|---|---|---|---|---|
| ⌊W2⌋ | 1.7 | 0.5 | 0.25 | | ⌊W2⌋ | **0.99** | 0.95 | 0.93 |
| ⌊MM⌋ | 2.2 | 0.9 | 0.53 | | ⌊MM⌋ | 0.98 | 0.90 | 0.87 |
| ⌊MM:R⌋ | **1.4** | **0.4** | **0.22** | | ⌊MM:R⌋ | **0.99** | **0.96** | **0.94** |
| ⌊ID⌋ | 31.2 | 4.26 | 2.06 | | ⌊ID⌋ | 0.00 | 0.00 | 0.00 |
| ⌊MM-B⌋ | 45.9 | 46.1 | 47.74 | | ⌊MM-B⌋ | 0.28 | -0.08 | -0.14 |
| ⌊C⌋ | 100 | 100 | 100 | | ⌊C⌋ | 0.03 | -0.14 | -0.20 |
| ⌊QC⌋ | 94.7 | ≫100 | ≫100 | | ⌊QC⌋ | 0.17 | -0.01 | 0.05 |

Table 3: $\mathcal{L}^2$-UVP (%, on the left) and $\cos \in [-1, 1]$ (on the right) metric values for transport maps $\mathbb{Q}_{\text{Cpkt}} \to \mathbb{P}_{\text{CelebA}}$ fitted by OT solvers on 3 developed CelebA64 $\mathbb{W}_2$ benchmarks.

To address this question, we evaluate the most promising solvers in the task of generative modeling for CelebA $64 \times 64$ images of faces. For comparison, we add ⌊QC⌋, which has good generative performance [19]. For each solver, we train a generative network $G_\alpha$ with ResNet architecture from [19] to map a 128-dimensional normal distribution $\mathbb{S}$ to the data distribution $\mathbb{Q}$. As the loss function for generator, we use $\mathbb{W}_2^2(\mathbb{P}_\alpha, \mathbb{Q}) = \mathbb{W}_2^2(G_\alpha\sharp\mathbb{S}, \mathbb{Q})$ estimated by each solver. We perform GAN-style training, where gradient updates of the generator alternate with gradient steps of OT solver (discriminator) (§B.2.3). We show sample generated images in the top row of each subplot of Figure 5 and report FID [13]. On the bottom row, we show the pushforward of the OT map from $\mathbb{P}_\alpha = G_\alpha\sharp\mathbb{S}$ to $\mathbb{Q}$ extracted from the OT solver. Since the model converged ($\mathbb{P}_\alpha \approx \mathbb{Q}$), the map should be nearly equal to the identity.

⌊W2⌋ provides the least quality (Figure 5a). This can be explained by the use of ConvICNN: the other solvers use convolutional architectures and work better. In general, the applicability of ICNNs to image-based tasks is questionable [16, §5.3] which might be a serious practical limitation.

⌊QC⌋ has strong generative performance (Figure 5b). However, as in §4.3-4.4, the recovered map is far from the identity. We suspect this solver has decent generative performance because it approximates some non-$\mathbb{W}_2^2$ dissimilarity measure in practice.

⌊MM⌋ results in a generative model that produces blurry images (Figure 5c). The computed transport map $\text{id}_{\mathbb{R}^D} - \nabla f_\theta$ is too far from the identity due to the gradient deviation. This leads to inaccurate gradient computation used to update the generator and explains why the generator struggles to improve. We emphasize that in §4.4 ⌊MM⌋ does not notably suffer from the gradient deviation. Probably, this is due to measures being absolutely continuous and supported on the entire $\mathbb{R}^D$. This is not the case in our generative modeling setup, where generated and data measures are supported on low-dimensional manifolds in $\mathbb{R}^D$.

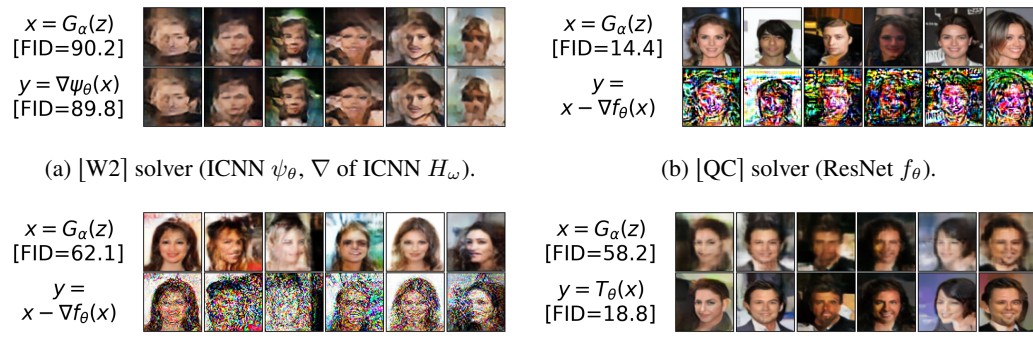

$x = G_\alpha(z)$
[FID=90.2]

$y = \nabla\psi_\theta(x)$
[FID=89.8]

(a) $\lfloor W2 \rfloor$ solver (ICNN $\psi_\theta$, $\nabla$ of ICNN $H_\omega$).

$x = G_\alpha(z)$
[FID=14.4]

$y = x - \nabla f_\theta(x)$

(b) $\lfloor QC \rfloor$ solver (ResNet $f_\theta$).

$x = G_\alpha(z)$
[FID=62.1]

$y = x - \nabla f_\theta(x)$

(c) $\lfloor MM \rfloor$ solver (ResNet $f_\theta$, UNet $H_\omega$).

$x = G_\alpha(z)$
[FID=58.2]

$y = T_\theta(x)$
[FID=18.8]

(d) $\lfloor MM:R \rfloor$ solver (UNet $T_\theta$, ResNet $g_\omega$).

Figure 5: Random images produced by trained generative models with OT solvers. The 1st line shows random generated images $x = G_\alpha(z) \sim \mathbb{P}_\alpha$, $z \sim \mathbb{S}$. The 2nd line shows computed transport map from the generated $x = G_\alpha(z) \sim \mathbb{P}_\alpha$ to the data distribution $\mathbb{Q}$.

Reversed $\lfloor MM:R \rfloor$ overcomes the problem of $\lfloor MM \rfloor$ with the gradient deviation but still leads to blurry images (Figure 5d). Interestingly, the fitted transport map $T_\theta$ significantly improves the quality and images $T_\theta \circ G_\alpha(z)$ are comparable to the ones with $\lfloor QC \rfloor$ solver (Figure 5b).

We emphasize that formulations from $\lfloor MM \rfloor$, $\lfloor MM:R \rfloor$ solvers are maximin: using them in GANs requires solving a challenging *min-max-min* optimization problem. To handle this, we use three nested loops and stochastic gradient descent-ascent-descent. In our experiments, the training was not stable and often diverged: the reported results use the best hyperparameters we found, although there may exist better ones. The difficulty in selecting hyperparameters and the unstable training process are limitations of these solvers that need to be addressed before using in practice.

## 5 Conclusion

Our methodology creates pairs of continuous measures with ground truth quadratic-cost optimal transport maps, filling the missing gap of benchmarking continuous OT solvers. This development allows us to evaluate the performance of quadratic-cost OT solvers in OT-related tasks. Beyond benchmarking the basic transport problem, our study of generative modeling reveals surprising patterns: bad OT solvers can yield good generative performance, and simply reversing asymmetric solvers can affect performance dramatically.

**Limitations.** We rely on ICNN gradients as $\mathbb{W}_2$ optimal transport maps to generate pairs of benchmark measures. It is unclear whether analogous constructions can be used for other costs such as $\mathbb{W}_1$. We also limit our benchmark pairs to be absolutely continuous measures while limiting the ground truth transport maps to be gradients of ICNNs, which may not have enough representational power. While we reveal a discrepancy between performance in OT-related tasks and performance in generative modeling, in-depth study is needed to answer questions such as what exact dissimilarity metric $\lfloor QC \rfloor$ implies that explains its generative performance while poorly approximating $\mathbb{W}_2$.

**Potential impact.** We expect our benchmark to become a standard benchmark for continuous optimal transport as part of the ongoing effort of advancing computational OT, in particular, in its application to generative modeling. As a result, we hope our work can improve the quality and reusability of OT-related research. One potential negative is that our benchmark might narrow the evaluation of future OT solvers to the datasets of our benchmark. To avoid this, besides particular benchmark datasets, in §3 we describe a generic method to produce new benchmark pairs.

ACKNOWLEDGEMENTS. The problem statement was developed in the framework of Skoltech-MIT NGP program. The work of Evgeny Burnaev was supported by the Ministry of Science and Higher Education of the Russian Federation grant No. 075-10-2021-068. The MIT Geometric Data Processing group acknowledges the generous support of Army Research Office grants W911NF2010168 and W911NF2110293, of Air Force Office of Scientific Research award FA9550-19-1-031, of National Science Foundation grants IIS-1838071 and CHS-1955697, from the CSAIL Systems that Learn program, from the MIT–IBM Watson AI Laboratory, from the Toyota–CSAIL Joint Research Center, from a gift from Adobe Systems, from an MIT.nano Immersion Lab/NCSOFT Gaming Program seed grant, and from the Skoltech–MIT Next Generation Program.

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
