# A  Benchmark Pairs Details

In Appendix A.1 we discuss the details of high-dimensional benchmark pairs. Appendix A.2 is devoted to Celeba $64 \times 64$ images benchmark pairs.

## A.1  High-dimensional Benchmark Pairs

The benchmark creation example is given in Figure 1. In each dimension we fix random Gaussian mixtures $\mathbb{P}, \mathbb{Q}_1, \mathbb{Q}_2$ (in the code we hard-code the random seeds) and use them to create a benchmark.

To generate a random mixture of $M$ Gaussian measures in dimension $D$, we use the following procedure. Let $\delta, \sigma > 0$ (we use $\delta = 1, \sigma = \frac{2}{5}$) and consider the $M$-dimensional grid

$$G = \{-\frac{\delta \cdot M}{2} + i \cdot \delta \text{ for } i = 1, 2, \dots, M\}^D \subset \mathbb{R}^D.$$

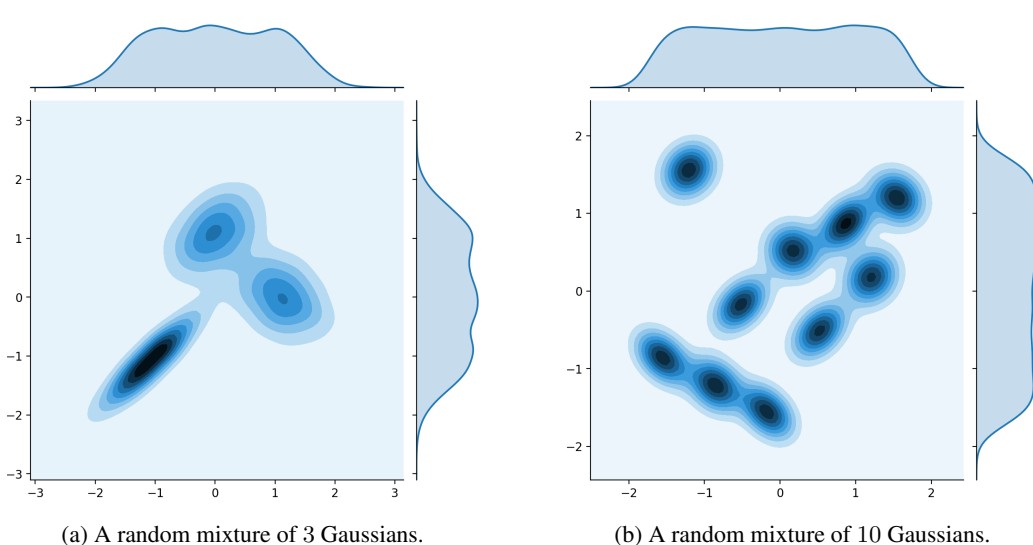

(a) A random mixture of 3 Gaussians.          (b) A random mixture of 10 Gaussians.

Figure 6: Randomly generated Gaussian mixtures. Projection on to first two dimensions.

We pick $M$ random points $\mu'_1, \dots \mu'_M \in G$ such that no pair of points has any shared coordinate. We initialize random $A'_1, \dots, A'_M \in \mathbb{R}^{D \times D}$, where each row of each matrix is randomly sampled from $D - 1$ dimensional sphere in $\mathbb{R}^D$. Let $\Sigma'_m = \sigma^2 \cdot (A'_m) \cdot (A'_m)^\top$ for $m = 1, 2, \dots, M$ and note that $[\Sigma'_m]_{dd} = \sigma^2$ for $d = 1, 2, \dots, D$. Next, we consider the Gaussian mixture $\frac{1}{M} \sum_{m=1}^{M} \mathcal{N}(\mu'_m, \Sigma'_m)$. Finally, we normalize the mixture to have axis-wise variance equal to 1, i.e. we consider the final mixture $\frac{1}{M} \sum_{m=1}^{M} \mathcal{N}(\mu_m, \Sigma_m)$, where $\mu_m = a\mu'_m$ and $\Sigma_m = a^2 \Sigma_m$. The value $a \in \mathbb{R}_+$ is given by

$$a^{-1} = \sqrt{\frac{\sum_{m=1}^{M} \|\mu'_m\|^2}{M \cdot D} + \sigma^2}.$$

Gaussian mixtures created by the procedure have $D$ same nice marginals, see Figure 6.

## A.2  CelebA $64 \times 64$ Images Benchmark Pairs

We fit 3 generative models on CelebA64 aligned faces dataset with a 128-dimensional latent Gaussian measure to sample from their distribution, using WGAN-QC [19] with a ResNet generator network. For trials $k = 1, 2$, we keep generator checkpoints after $1000, 5000, 10000$ iterations to produce measures $\mathbb{Q}^k_{\text{Early}}, \mathbb{Q}^k_{\text{Mid}}, \mathbb{Q}^k_{\text{Late}}$ respectively. In the last trial $k = 3$, we keep only the final generator network checkpoint after $50000$ iterations which produces measure $\mathbb{P}^3_{\text{Final}}$. To make each of measures absolutely continuous, we add white Normal noise (axis-wise $\sigma = 0.01$) to the generators' output.

We use the generated measures to construct images benchmark pairs according to the pipeline described in §4.1. We visualize the pipeline in Figure 2.

# B Experimental Details

In Appendix B.1, we discuss the neural network architectures we used in experiments. All the other training hyperparameters are given in Appendix B.2.

## B.1 Neural Network Architectures

In Table 4 below, we list all the neural network architectures we use in continuous OT solvers. In every experiment we pre-train networks to satisfy $\nabla\psi_\theta(x) = x - \nabla f_\theta(x) \approx x$ and $H_\omega(y) \approx y$ at the start of the optimization. We empirically noted that such a strategy leads to more stable optimization.

| Solver | High-dimensional benchmark | CelebA benchmark | CelebA image generation |
|---|---|---|---|
| [LS] | $\psi_\theta, \phi_\omega : \mathbb{R}^D \to \mathbb{R}$ - DenseICNN (U) | N/A | |
| [MM-B] | $\psi_\theta : \mathbb{R}^D \to \mathbb{R}$ - DenseICNN (U) | $f_\theta : \mathbb{R}^D \to \mathbb{R}$ - ResNet | |
| [QC] | $\psi_\theta : \mathbb{R}^D \to \mathbb{R}$ - DenseICNN (U) | $f_\theta : \mathbb{R}^D \to \mathbb{R}$ - ResNet | |
| [MM] | $\psi_\theta : \mathbb{R}^D \to \mathbb{R}$ - DenseICNN (U) | $f_\theta : \mathbb{R}^D \to \mathbb{R}$ - ResNet | |
| | $H_\omega : \mathbb{R}^D \to \mathbb{R}^D$ - $\nabla$ of DenseICNN (U) | $H_\omega : \mathbb{R}^D \to \mathbb{R}^D$ - UNet | |
| [MM:R] | $T_\theta : \mathbb{R}^D \to \mathbb{R}^D$ - $\nabla$ of DenseICNN (U) | $T_\theta : \mathbb{R}^D \to \mathbb{R}^D$ - UNet | |
| | $\phi_\omega : \mathbb{R}^D \to \mathbb{R}$ - DenseICNN (U) | $g_\omega : \mathbb{R}^D \to \mathbb{R}$ - ResNet | |
| [MMv1] | $\psi_\theta : \mathbb{R}^D \to \mathbb{R}$ - DenseICNN | N/A | |
| [MMv2] | $\psi_\theta : \mathbb{R}^D \to \mathbb{R}$ - DenseICNN | $\psi_\theta : \mathbb{R}^D \to \mathbb{R}$ - ConvICNN64 | |
| [W2] | $H_\omega : \mathbb{R}^D \to \mathbb{R}^D$ - $\nabla$ of DenseICNN | $H_\omega : \mathbb{R}^D \to \mathbb{R}^D$ - $\nabla$ of ConvICNN64 | |
| [MMv2:R] | $T_\theta : \mathbb{R}^D \to \mathbb{R}^D$ - $\nabla$ of DenseICNN | $T_\theta : \mathbb{R}^D \to \mathbb{R}^D$ - $\nabla$ of ConvICNN64 | |
| [W2:R] | $\phi_\omega : \mathbb{R}^D \to \mathbb{R}$ - DenseICNN | $\phi_\omega : \mathbb{R}^D \to \mathbb{R}$ - ConvICNN64 | |

Table 4: Network architectures we use to parametrize potential $f$ (or $\psi$) and map $H$ in tested solvers. In the reversed solvers we parametrize second potential $g$ (or $\phi$) and forward transport map $T$ by neural networks.

In the **high-dimensional benchmark**, we use DenseICNN architecture from [16, §B.2]. It is a fully-connected neural net with additional input-quadratic skip-connections. This architecture can be made input-convex by limiting certain weights to be non-negative. We impose such as a restriction only for [MMv1],[MMv2],[W2] solvers which require networks to be input-convex. In other cases, the network has no restrictions on weights and we denote the architecture by DenseICNN (U). In experiments, we use the implementation of DenseICNN from the official repository of [W2] solver

https://github.com/iamalexkorotin/Wasserstein2GenerativeNetworks

More precisely, in the experiments with probability measures on $\mathbb{R}^D$, we use

$$\text{DenseICNN}[1; \max(2D, 64), \max(2D, 64), \max(D, 32)].$$

Here 1 is the rank of the input-quadratic skip connections and the other values define sizes of fully-connected layers the sequential part of the network. The notation follows [16, §B.2].

We emphasize that DenseICNN architecture $\psi_\theta$ has diffirentiable CELU [4] activation functions. Thus, $\nabla\psi_\theta$ is well-defined. In particular, artificial $\beta \cdot \|x\|^2/2$ for $\beta = 10^{-4}$ is added to the output of the last layer of the ICNN. This makes $\psi_\theta$ to be $\beta$-strongly convex. As the consequence, $\nabla\psi_\theta$ is a bijective function with Lipschitz constant lower bounded by $\beta$, see the discussion in [16, §B.1].

In the **experiments with CelebA images**, for parametrizing the potential $f = f_\theta : \mathbb{R}^D \to \mathbb{R}$ in [MM], [QC], [MM-B], we use ResNet architecture from the official WGAN-QC [19] repository:

https://github.com/harryliew/WGAN-QC

To parametrize the map $H = H_\omega : \mathbb{R}^D \to \mathbb{R}^D$ in [MM] solver, we use UNet architecture from

https://github.com/milesial/Pytorch-UNet

In [MMv2], [W2] solvers we parametrize $\psi = \psi_\theta$ and $H = H_\omega = \nabla\phi_\omega$, where both $\psi_\theta, \phi_\omega$ have ConvICNN64 architecture, see Figure 7. We artificially add $\beta \cdot \|x\|^2/2$ (for $\beta = 10^{-4}$) to the output of the output of the ConvICNN64 to make its gradient bijective.

In the architecture, *PosConv2D* layers are usual 2D convolutional layers with all weights (except biases) restricted to be non-negative. *Conv2D-CQ* (convex quadratic) are fully convolutional blocks

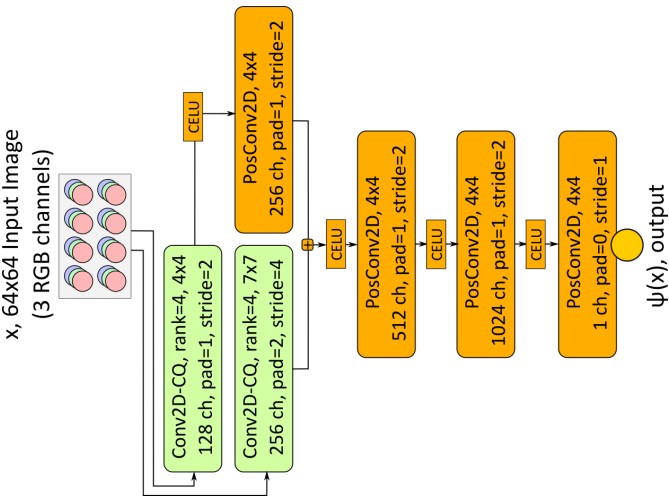

Figure 7: Convolutional ICNN architecture we use for processing $64 \times 64$ RGB images.

which output a tensor whose elements are input-quadratic functions of the input tensor. In Figure 8, we present the architecture of Conv2D-CQ block. Here, *GroupChannelSumPool* operation corresponds to splitting the tensor per channel dimension into $n_{out}$ sequential sub-tensors (each of $r$ channels) and collapsing each sub-tensor into one 1-channel tensor by summing $r$ channel maps. The layer can be viewed as the convolutional analog of *ConvexQuadratic* dense layer proposed by [16, §B.2].

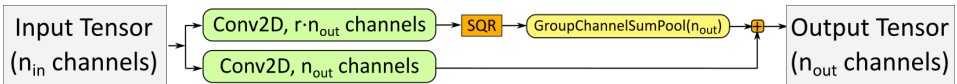

Figure 8: 2D convolutional convex quadratic block.

In the **CelebA image generation experiments**, we also use ResNet architecture for the generator network $g$. The implementation is taken from WGAN-QC repository mentioned above.

## B.2 Hyperparameters and Implementation Details

The evaluation of all the considered continuous solvers for evaluation is not trivial for two reasons. First, not all the solvers have available user-friendly *Python* implementations. Next, some solvers are not used outside the GAN setting. Thus, for considering them in the benchmark, proper extraction of the $\mathbb{W}_2$ solver (discriminator part) from the GAN is needed.

We implement most of the solvers from scratch. In all the cases, we use Adam optimizer [15] with default hyperparameters (exept the learning rate). For solvers ⌊QC⌋ by [19] and ⌊W2⌋ by [16] we use the code provided by the authors in the official papers' GitHub repositories.

### B.2.1 High-dimensional Benchmark Pairs

We report the hyper parameters we use in high-dimensional benchmark in Table 5. *Total iterations* column corresponds to optimizing the potential $f_\theta$ (or $\psi_\theta$) to *maximize* the dual form (8). In maximin solvers, there is also an inner cycle which corresponds to solving the inner *minimization* problem in (8). The hyperparameters are chosen empirically to best suit the considered evaluation setting.

For ⌊QC⌋ solver large batch sizes are computationally infeasible since it requires solving a linear program at each optimization step [19, §3.1]. Thus, we use batch size $64$ as in the original paper. ⌊W2⌋ solver is used with the same hyperparameters in training/evaluation of the benchmarks.

### B.2.2 CelebA $64 \times 64$ Images Benchmark Pairs

For the images benchmark, we list the hyperparameters in Table 6.

| Solver | Batch Size | Total Iterations | LR | Note |
|---|---|---|---|---|
| ⌊LS⌉ | 1024 | 100000 | $10^{-3}$ | Quadratic regularization with $\epsilon = 3 \cdot 10^{-2}$, see [36, Eq. (7)] |
| ⌊MM-B⌉ | 1024 | 100000 | $10^{-3}$ | None |
| ⌊QC⌉ | 64 | 100000 | $10^{-3}$ | OT regularization with $K = 1, \gamma = 0.1$, see [19, Eq. (10)] |
| ⌊MMv1⌉ | 1024 | 20000 | $10^{-3}$ | 1000 gradient iterations ($lr = 0.3$) to compute argmin in (8), see [39, §6]. Early stop when gradient norm $< 10^{-3}$. |
| ⌊MM⌉,⌊MMv2⌉ | 1024 | 50000 | $10^{-3}$ | 15 inner cycle iterations to update $H_\omega$, ($K = 15$ in the notation of [26, Algorithm 1]) |
| ⌊W2⌉ | 1024 | 250000 | $10^{-3}$ | Cycle-consistency regularization, $\lambda = D$, see [16, Algorithm 1] |

Table 5: Hyperparameters of solvers we use in high-dimensional benchmark. Reversed are not presdented in this table: they use the same hyperparameters as their original versions.

| Solver | Batch Size | Total Iterations | LR | Note |
|---|---|---|---|---|
| ⌊MM-B⌉ | 64 | 20000 | $3 \cdot 10^{-4}$ | None |
| ⌊QC⌉ | 64 | 20000 | $3 \cdot 10^{-4}$ | OT regularization with $K = 1, \gamma = 0.1$, see [19, Eq. (10)] |
| ⌊MM⌉ | 64 | 50000 | $3 \cdot 10^{-4}$ | 5 inner cycle iterations to update $H_\omega$, ($K = 5$ in the notation of [26, Algorithm 1]) |
| ⌊W2⌉ | 64 | 50000 | $3 \cdot 10^{-4}$ | Cycle-consistency regularization, $\lambda = 10^4$, see [16, Algorithm 1] |

Table 6: Hyperparameters of solvers we use in CelebA images benchmark.

### B.2.3 CelebA $64 \times 64$ Images Generation Experiment

To train a generative model, we use GAN-style training: generator network $G_\alpha$ updates are alternating with OT solver's updates (discriminator's update). The learning rate for the generator network is $3 \cdot 10^{-4}$ and the total number of generator iterations is $50000$.

In ⌊QC⌉ solver we use the code by the authors: there is one gradient update of OT solver per generator update. In all the rest methods, we alternate $1$ generator update with $10$ updates of OT solver (*iterations* in notation of Table 6). All the rest hyperparameters match the previous experiment.

The generator's gradient w.r.t. parameters $\alpha$ on a mini-batch $z_1, \ldots, z_N \sim \mathbb{S}$ is given by

$$\partial \mathbb{W}_2^2(\mathbb{P}_\alpha, \mathbb{Q})/\partial \alpha = \int_z \mathbf{J}_\alpha G_\alpha(z)^T \nabla f^*\big(G_\alpha(z)\big) d\mathbb{S}(z) \approx \frac{1}{N} \sum_{n=1}^{N} \mathbf{J}_\alpha G_\alpha(z_n)^T \nabla f_\theta\big(G_\alpha(z_n)\big) \quad (10)$$

where $\mathbb{S}$ is the latent space measure and $f_\theta$ is the current potential (discriminator) of OT solver. Note that in ⌊MM:R⌉ potential $f$ is not computed but the forward OT map $T_\theta$ is parametrized instead. In this case, we estimate the gradient (10) on a mini-batch by $\frac{1}{N} \sum_{n=1}^{N} \mathbf{J}_\alpha G_\alpha(z_n)^T \big(\mathrm{id}_{\mathbb{R}^D} - T_\theta\big)$.