# OpenReview forum: "Do Neural Optimal Transport Solvers Work? A Continuous Wasserstein-2 Benchmark"
_NeurIPS.cc/2021/Conference — NeurIPS 2021 Poster_

### Official Review · Reviewer_oJwC · 2021-07-16

**Rating:** 5
**Confidence:** 5

**Summary:**

This work introduces a continuous Wasserstein-2 benchmark to assess the qualities of different neural optimal transport solvers. The work uses input convex neural networks (ICNN) to construct pairs of measures whose ground truth OT maps can be obtained analytically. This strategy yields pairs of continuous benchmark measures in high-dimensional spaces such as spaces of images. The authors thoroughly evaluate existing optimal transport solvers. The study reveals crucial limitations of existing solvers.

**Limitations And Societal Impact:**

There are more limitations of the current design:

1. Secondary boundary condition should be emphasized, which significantly affect the computational quality.
2. The regularity of the OT maps need to be carefully considered, if the target measure support is non-convex, the OT maps can not be represented by deep neural networks. This fact should be carefully addressed.

**Main Review:**

Optimal transportation maps become more and more important and popular in deep learning field. Due to its highly non-linear nature, reliable computation with high precision is challenging. So far, there is no solid benchmark to assess different solvers. This work proposes a benchmark to tackle this problem. Hence it has importance for deep learning fields.

The manuscript is well written. The introduction part is highly motivated and encouraging. The theoretic foundation part is clearly written, easy to follow, but can be further improved.

1. The background on optimal transport is insufficient. There are many existing methods based on Brenier theorem and solving Monge-Ampere equations. For example, Benamou-Brenier used fluid dynamic formulation to solve L2 optimal transportation map; Yau et al give the intrinsic connection between convex geometry and optimal transport, and solve the OT map problem using Alexandrov formulation, see the work  of Lei et al "A Geometric Understanding of Deep Learning",   Engineering, Volume 6, Issue 3, March 2020, Pages 361-374. The algorithm there has theoretic foundation.

2. The benchmark ignores one important aspect of optimal transportation map: the boundary condition. In theory, the image of the OT map of the source domain should equal to the target domain, this is the most difficult constraint to satisfy in practice and affect the computational result crucially. According to Figalli's works, if the target domain is non-convex, then the OT map itself may not be continuous, therefore cannot be represented by the deep nerual networks. Even for the benchmark itself, the ground truth map may not be representable by the network. According to Yau's works, this is the reason for mode collapsing. Hence this need to be emphasized and carefully designed in the bench mark.

The experimental results are convincing, different solvers are compared.

3. It will be more helpful to add geometric based approaches for the test. It will be more convincing if the convergence error is also added. According to the theoretic analysis, the Brenier potential error is O(h^2) where h is the diameter of the cells in the tessellation.

In general, the idea of Benchmark is important, it will make the whole field more rigorous. Current design can be further improved by emphasizing the 2nd boundary condition, and considering the regularity of the OT maps.


**Time Spent Reviewing:**

3

---

> ### Author Response · Authors · 2021-08-07
> **Answers to Reviewer oJwC**
>
> Dear reviewer,
>
> Thanks for your thoughtful review. Before providing responses to your individual points we wish to clarify that our paper does not propose a new OT solver and is not intended as a survey of computational OT.  Rather, it is a benchmark paper: we propose a methodology and datasets to evaluate OT solvers in the continuous setting and test existing solvers.
>
> Please find  answers to your comments and questions below.
>
> **(1) Background on OT and geometric methods.**
>
> Although we are happy to extend our exposition of previous work, we underscore that our paper is not intended as a survey of all OT solvers but rather as a benchmark for continuous dual OT solvers---the most prevalent approach to computational OT in recent machine learning research. Our current Section 2 mentions only existing solvers for continuous measures $\mathbb{P},\mathbb{Q}$.
>
> We appreciate the reference to the geometric solver by Lei et al. (2020) and can include a citation in our final draft. This solver, however, is outside the scope of our benchmark: It is not a continuous but semi-discrete solver, i.e., $\mathbb{Q}$ is a discrete measure with fixed support. Their solver is inconsistent with our setup and designed for a different setting, and hence we are unable to add it to our evaluation.
>
> **(2) Regularity of OT maps.**
>
> In our benchmark, we construct $\mathbb{P}$ to be continuous and supported on $\mathbb{R}^{D}$ and ICNNs $\psi_{\theta}$ to have bijective gradient (see lines 181-183 and Appendix B.1). In this case, $\nabla\psi_{\theta}\sharp\mathbb{P}$ is also continuous and supported on $\mathbb{R}^{D}$. This ensures existence of bijective inverse OT map $(\nabla\psi_{\theta})^{-1}$ that can be used used for benchmarking; we used this construction in creating image benchmark pairs (Section 4.4). Lines 300-302 mention this property of our constructed benchmark pairs.
>
> Our generic methodology can be easily used to generate benchmark measures supported on non-convex sets and having discontinuous OT maps. Our key idea (line 167-171) is not limited to ICNNs with continuous gradients or measures $\mathbb{P}$ with convex supports.
>
> **(3) Second boundary condition.**
>
> If we correctly understand the comment, *second boundary condition* refers to the condition $T\sharp\mathbb{P}=\mathbb{Q}$. In the beginning of Section 2 (lines 110-115), we explain that for evaluation we consider dual form solvers, which---in contrast to primal-form solvers---avoid direct imposition of this condition. After recovering $\hat{T}\approx T^{*}$ from the dual problem, we evaluate $\mathcal{L}^{2}$-UVP metric. It not only checks how good is the fitted map to the true one w.r.t. the $\mathcal{L}^{2}(\mathbb{P})$ norm, but also upper bounds the closeness of $\hat{T}\sharp\mathbb{P}$ to $\mathbb{Q}$ in terms of $\mathbb{W}_{2}^{2}$ distance, i.e., how the second boundary condition is satisfied, see [17, eq. (17)]. Therefore, our benchmark metric also examines the second boundary condition.
>
> **Concluding remarks.**
> Please respond to our post to let us know if the clarifications above suitably address your concerns about our work.  We are happy to address any remaining points during the discussion phase; if the responses above are sufficient, we kindly ask that you consider raising your score.

---

> > ### Author Response · Authors · 2021-08-31
> > **Looking forward to your final feedback**
> >
> > Dear Reviewer oJwC,
> >
> > We thank you for your review and appreciate your time reviewing our paper.
> >
> > The end of the rebuttal phase is approaching. We would be grateful if we could hear your feedback regarding our answers to the reviews. We are happy to address any remaining points during the remaining period.
> >
> > Thanks in advance,
> >
> > Paper4129 authors

---

### Official Review · Reviewer_paeq · 2021-07-16

**Rating:** 7
**Confidence:** 4

**Summary:**

The authors present a continuous Wasserstein-2 benchmark using input convex neural networks to obtain ground truth continuous OT maps. They evaluate continuous OT solvers based on how well they reproduce the ground truth transport map, how well they approximate the derivative of the potential, and how well they perform in a GAN framework. This provides a basis to evaluate current and future continuous 2-Wasserstein solvers.

**Limitations And Societal Impact:**

Yes

**Main Review:**

Overall this was an excellent read and I found the use of the ICNN to generate ground truth OT maps novel and clever. The authors mention "Because of the choice of [W2], subsequent evaluation might slightly favor ICNN-based methods" (Line 177) and elaborate in the limitations section. This seems like a major limitation, the fact that the transport maps are not based on any real distribution and are in fact sampled from the set of potentials parameterized by ICNNs is highly biased. Would it be possible to examine this bias in some way? Perhaps some P, Q pairs with tractable transport maps that do not require ICNNs?

The evaluation of W2 solvers in generative modeling is also extremely useful. While this study has a number of limitations it is well posed and a useful benchmark for future work in W_2 solvers. I found this a fascinating read and applaud the authors for their efforts.

------------------------------
Edit post response:
I am satisfied by the authors' response. I keep my score.

I initially had the same understanding as reviewer oJwC made in his point 2. Given that this is a mainly a benchmark paper, the construction of the benchmark is of utmost importance. It would improve the paper to further emphasis the construction of the ground truth based on maps model-able by ICNNs, which is the most significant limitation of this work. Having a clear discussion of the use of DenseICNN parameterizing the ground truth transport maps earlier in this work perhaps in section 3 (rather than just in the limitations paragraph at the end) would both make it clearer how the benchmarks are constructed and what the limitations are. The paragraph titled "Arbitrary pairs"on Line 172 is misleading as a ground truth map is not constructed between arbitrary pairs, but between P and something close to Q based on how well the ICNN is fit; I suggest "Approximating arbitrary pairs" or similar would be clearer.

 While I agree other convex parameterizations could be used, this is an important point and deserves further clarification.

**Time Spent Reviewing:**

4

---

> ### Author Response · Authors · 2021-08-07
> **Answers to Reviewer paeq**
>
> Dear reviewer,
>
> Thanks for your thoughtful review. We appreciate that you are positive about our results and recognize our efforts to bring transparency, replicability, and healthy competition to modern OT research. Please find below the answers to your comments and questions.
>
> **(1) Other parametrization and ICNN bias**.
>
> As noted in lines 102-103, for the continuous case, the exact OT solution is only known for a small number of pairs of measures; our ICNN parameterization already contains several of these cases exactly. For instance, for the Gaussian/location-scatter cases, ICNN does not have bias: the DenseICNN we use can represent the quadratic potentials due to the quadratic skip-connections that we employ.
>
> Our work represents a significant step in extending the class of pairs of measures with analytically-known OT solution. Our methodology is generic and is not limited to ICNN parametrization: any other parameterization of convex functions can also be used.

---

### Official Review · Reviewer_jpfi · 2021-07-17

**Rating:** 6
**Confidence:** 3

**Summary:**

This paper proposes a continuous Wasserstein-2 benchmark to evaluate different neural OT solvers including WGAN varients. Extensive experiments are performed on toy datasets and a real face dataset.

**Limitations And Societal Impact:**

The authors have addressed some limitations.

**Main Review:**

Strength:
1. It's good to see a benchmark that evaluates different neural OT solvers. This could benefit future OT application works that choose the best OT solver.
2. Comprehensive OT solvers are investigated, including some WGAN variants that can be potentially used as OT solvers.
3. Various metrics such as L^2-UVP, cos, and FID are used to evaluate OT solvers.

Weakness:

I think two important theoretical problems remain unsolved in this paper:

1. In line 216, the authors mentioned that to compute the L^2-UVP and the cos metrics, they use 2^14 random samples from P. They actually computed the estimated values for the two metrics.
I would like to know the sample complexity of estimating the two metrics, i.e., the estimation error of L^2-UVP and cos in terms of the number of samples required. Is there a curse of dimensionality problem in estimating the L^2-UVP and cos,
just like computing the Wasserstein distance for two continuous distributions using sampling methods? This is important because if the number of required samples grows exponentially as the dimensionality increases,
then the metrics may not be accurate in evaluating OT solvers in the 64d space but only with 2^14 samples.

2. In line 175, the authors construct a benchmark pair (P, 1/2 (\nabla \psi_1 + \nabla \psi_2)#P), and the 1/2 (\nabla \psi_1 + \nabla \psi_2) is used as the ground truth transport map if I understand correctly.
My concern is that "is 1/2 (\nabla \psi_1 + \nabla \psi_2) really the optimal transport map between P and 1/2 (\nabla \psi_1 + \nabla \psi_2)#P"? Remember that \psi_1 and \psi_2 are approximated by the W2 solver.
I think this paper should theoretically show "how far is 1/2 (\nabla \psi_1 + \nabla \psi_2) away from the real optimal transport map between P and 1/2 (\nabla \psi_1 + \nabla \psi_2)#P", and show how this
gap affects the metrics that are used to evaluate OT solvers.

**Time Spent Reviewing:**

5

---

> ### Author Response · Authors · 2021-08-07
> **Answers to Reviewer jpfi**
>
> Dear reviewer,
>
> Thanks for your thoughtful review. Please find below the answers to your comments and questions.
>
> **(1) Sample complexity of metrics**.
>
> Statistical questions about our error metrics are beyond the scope of our benchmark. Both $\mathcal{L}^{2}\text{-UVP}$ and $\cos$ metrics admit straightforward Monte Carlo estimation procedures. The well-known sample complexity of Monte-Carlo integration grows as $O(n^{-1/2})$ with $n$ samples, which does not suffer from the curse of dimensionality.  For $\mathcal{L}^{2}\text{-UVP}$, the denominator is a constant for comparing methods, so the relevant sample complexity remains $O(n^{-1/2})$.
>
> Although a theoretical variance bound may be out of reach, following standard empirical practice in statistics, we can easily verify the reliability of our estimator by evaluating it multiple times with different random seeds and providing confidence intervals; we can easily include these estimates in our final version of the paper.  Empirically, the fluctuation of $\mathcal{L}^{2}\text{-UVP}$ [$\cos$] metric on different batches typically does not exceed $0.5-1$% [$0.02$-$0.03$]; the metric decreases [increases, resp.] during training, as expected.
>
> Our experiments show that our metrics are reliable and helpful to distinguish good/bad solvers, which can be seen, for example, from qualitative results of Figure 2a agreeing with metrics of Table 3. We additionally point out that, for example, $\mathcal{L}^{2}$-UVP metric is already in use in other OT papers [16, 17].
>
> **(2) Averaging benchmarks.**
>
> If $\psi_1$ and $\psi_2$ are both convex, then $\frac{1}{2}(\psi_1 + \psi_2)$ is convex. Following lines 167-171 and 178-180, the map $\frac{1}{2}(\nabla\psi_{1}+\nabla\psi_{2}) = \nabla (\frac{1}{2}(\psi_1 + \psi_2))$ is then the exact OT map between $\mathbb{P}$ and $\frac{1}{2}(\nabla\psi_{1}+\nabla\psi_{2})\sharp\mathbb{P}$.
>
> **Concluding remarks.**
> Please respond to our post to let us know if the clarifications above suitably address your concerns about our work.  We are happy to address any remaining points during the discussion phase; if the responses above are sufficient, we kindly ask that you consider raising your score.
>
> Our benchmark fills an important niche in computational optimal transport, since most recent OT papers use a restricted set of self-generated ad hoc examples to test their methods.  The current scenario makes it hard to compare methods or accurately assess the value of different methods for computational OT. Our work brings a convenient standardized way to assess OT algorithms and as a result will encourage healthy competition in this developing subfield of ML.

---

> > ### Comment · Reviewer_jpfi · 2021-08-29
> > **Response**
> >
> > Thank you for your response to my questions! I think you answered my questions and thus I'm willing to increase my score!
> >
> > Thanks!
> > jpfi

---

### Author Response · Authors · 2021-08-26
**Looking forward to your final feedback**

Dear Reviewers,

We thank you for your review and appreciate your time reviewing our paper.

The end of the discussion period is close. We would be grateful if we could hear your feedback regarding our answers to the reviews. We are happy to address any remaining points during the remaining discussion period.

Thanks in advance,

Paper4129 authors

---

### Decision · Program_Chairs · 2021-09-27

**Decision:**

Accept (Poster)

**Comment:**

In this paper, the authors propose an interesting approach to evaluating transport plans. There exist some concerns about methodology. However, to my knowledge, I have never seen this type of study before in OT problems. So, the problem itself is new and novel. Thus, I also would like to vote for acceptance. For the camera-ready version, I expect the authors to revise the paper based on the reviewer's comments.